# TOWARDS THE DECISIVE FACTOR OF SYMBOLIC GENERALIZATION OF DNNS

## ABSTRACT

The decisive factor that drives deep neural networks (DNNs) to learn non-generalizable representations (*i.e.,* non-generalizable interactions between input variables) has been a persistent challenge in the field of symbolic generalization. In this paper, we quantify the generalization power of such interactions encoded by DNNs, and we discover that DNNs usually learn non-generalizable interactions from a few samples, referred to as *confusing samples*. The emergence of confusing samples during the training process explains the overfitting of a DNN. We further discover that the composition of confusing samples is determined by the randomness of parameter initialization in the low layers of a DNN. In contrast, other factors, such as high-layer parameters and network architecture, have much less impact on the composition of confusing samples. Consequently, two DNNs initialized with different low-layer parameters will eventually learn entirely different sets of confusing samples, even though they have similar performance.

## 1 INTRODUCTION

Although the generalization power of deep neural networks (DNNs) has received significant attention (Petrini et al., 2022; Boopathy et al., 2023; Foret et al., 2021; Nikolikj et al., 2024), the underlying mechanisms behind generalization remain poorly understood. To address this gap, a new trend is to explore more fine-grained explanations of generalization.

Therefore, a growing line of work, namely **symbolic generalization** (Ren et al., 2024a;b; Zhou et al., 2024; Li & Zhang, 2023b) has received increasing attention, which explains a DNN's generalization power through the generalizability of its encoded inference patterns. Thus, this paper explores a fundamental issue in symbolic generalization: *why does a DNN mainly use non-generalizable inference patterns to classify certain samples, yet predominantly use generalizable ones for others?*

**Background.** The emerging direction of symbolic generalization has provided a new explanation strategy (see survey in Appendix A), *i.e., mathematically proving that the intricate inference patterns used by a DNN for classification can be accurately explained by a small set of interactions (Ren et al., 2024a).* Each interaction represents an AND relationship or an OR relationship between input variables. For example, as Figure 1 shows, the co-appearance of words/tokens in $S = \{red, apple, falls\}$ in the input prompt triggers an AND interaction $S$ encoded by the DNN, and it contributes a numerical effect to boost the confidence of generating "*gravity*." It is proven by (Ren et al., 2024a) that numerical effects of a small number of interactions can accurately mimic the DNN's outputs on exponentially many input samples. This property has been widely considered (Ren et al., 2024a; Chen et al., 2024; Ren et al., 2024b; Zhou et al., 2024; Li & Zhang, 2023b) to theoretically guarantee faithfulness of taking interactions as primitive inference patterns in a DNN. Please see the **video demo** in the supplementary material for symbolic generalization.

However, pioneering studies in symbolic generalization have encountered difficulties in addressing the second critical question, *i.e.,* how to rigorously explain the generalizability of interactions in order to analyze the performance of DNNs. To this end, prior studies have mainly explored the generalizability (Ren et al., 2024b; Zhou et al., 2024) and robustness (Liu et al., 2023) of interactions in an empirical manner. However, one essential factor behind these studies remains unexplored: **why do certain samples lead DNNs to encode non-generalizable interactions, while other samples mainly induce generalizable ones? This question has hampered the field for years.** Specifically, an interaction extracted from a training sample is considered generalizable if it is effectively used by

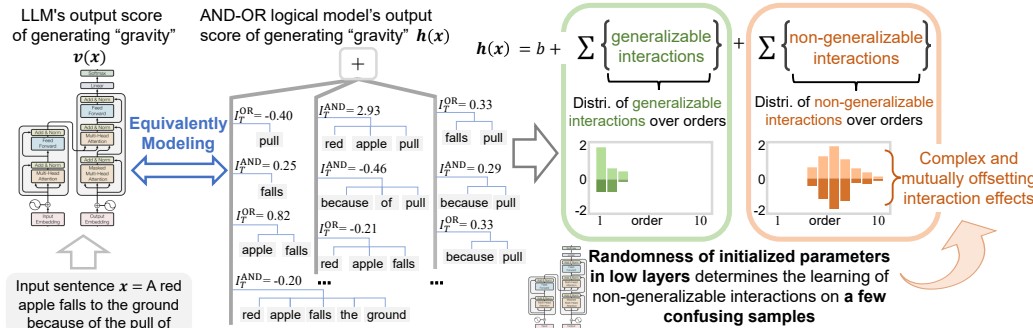

Figure 1: (Left) It has been proven (Ren et al., 2024a) that the intricate inference patterns of a DNN on an input sample can be faithfully represented in mathematics by a logical model consisting of AND-OR interactions. **Please see Figure 11 for examples of an AND-OR logical model that explains LLMs.** (Right) Some interactions can generalize to testing samples, but others cannot. We find that the randomness of initialized parameters in low layers leads different DNNs to learn non-generalizable interactions from entirely different sets of samples, referred to as *confusing samples*.

the DNN to classify testing samples. Addressing this question would reveal the essence of overfitting at a finer granularity.

**Our work.** Considering the above technical challenge, we have discovered a counterintuitive conclusion: training samples are not the primary factor in determining whether a DNN learns generalizable or non-generalizable interactions. In contrast, *the randomness of initialized parameters in the low layers plays a far more dominant role in determining the learning of non-generalizable interactions*, provided that the quality of different training datasets is not drastically different. First, we find that non-generalizable interactions do not appear uniformly across different samples. As a DNN gradually overfits during training, **it mainly learns non-generalizable interactions from a small subset of input samples, which we define as confusing samples.** These non-generalizable interactions are typically complex and have mutually offsetting interaction effects. In contrast, the majority of other samples (*i.e.,* non-confusing samples) mainly contain simple interactions, which generalize well to testing samples.

Then, we find that **the randomness of initialized parameters in low layers is the key factor that determines the composition of a DNN's confusing samples.** In comparison, other factors such as parameter initialization in high layers and network architecture have much less impact. Different DNNs often produce entirely distinct sets of confusing samples, even when they have the same architecture and similar performance. However, if two DNNs share the same low-layer parameters, they will also have the same composition of confusing samples, even when they have fully different parameter initializations in high layers, architectures, and training datasets.

**Application potential.** Our finding that the initialization of low-layer parameters is the decisive factor for confusing samples provides new guidance for training DNNs. Prior studies have often mistakenly attributed overfitting on certain samples to the quality of the training data, which may misrepresent the true cause. By contrast, our discovery highlights the importance of tailoring data valuation to each specific DNN, rather than expecting a certain set of "high-quality" samples to universally benefit all models. Moreover, follow-up studies shown in Appendix D further demonstrate that eliminating non-generalizable interactions (*i.e.,* reducing the number of confusing samples) during the DNNs' training process can slightly enhance the performance of DNNs during training.

## 2 CONFUSING SAMPLES WITH NON-GENERALIZABLE INTERACTIONS

### 2.1 PRELIMINARIES: INTERACTIONS

Given a DNN $v$ and an input sample $\boldsymbol{x} = [x_1, x_2, \ldots, x_n]^T$ with $n$ input variables[1], indexed by $N = \{1, 2, \ldots, n\}$. Let $v(\boldsymbol{x}) \in \mathbb{R}$ denote a scalar output of the DNN, *e.g.,* the widely-used scalar

---

[1]In image classification, the input variables can be set as different patches of an image. In language generation, the embedding vector of a token can be considered as an input variable.

classification confidence in multi-category classification (Deng et al., 2022), as follows:

$$v(\boldsymbol{x}) \stackrel{\text{def}}{=} \log \frac{p(y = y^* | \boldsymbol{x})}{1 - p(y = y^* | \boldsymbol{x})} \in \mathbb{R}, \tag{1}$$

where $p(y = y^* | \boldsymbol{x})$ denotes the probability of classifying the sample $\boldsymbol{x}$ to the ground-truth label $y^*$.

**Problem setting:** A central challenge in *symbolic generalization* is to explain the intricate inference patterns of a DNN. To address this, recent advancements in symbolic generalization (Li & Zhang, 2023b; Ren et al., 2023a; 2024a) have proposed to use a logical model $h$ to explain the intricate inference patterns of a DNN $v$, whose mathematical faithfulness is theoretically guaranteed by the following two requirements. **(1) Fidelity requirement:** the logical model $h$ must accurately predict the network outputs $v(x)$ over a vast number of samples in $\Psi$. **(2) Conciseness requirement:** the logical model $h$ should encode sufficiently simple logic, so that the explanation remains transparent and easy to interpret, as follows.

$$\forall \boldsymbol{x}' \in \Psi, \quad h(\boldsymbol{x}') = v(\boldsymbol{x}'), \quad \text{subject to} \quad \text{complexity}(h) \leq M, \tag{2}$$

where $M$ is an upper bound on the complexity of the logical model $h$.

**The logical model $h$ is implemented as the following model comprising a set of AND-OR interactions (see Figure 1).** *Please refer to the **video demo** attached to the supplementary material, which illustrates the central idea of explanation theory. Please see Figure 11 for examples of an AND-OR logical model that explains LLMs.*

$$\forall \boldsymbol{x}' \in \Psi, \quad h(\boldsymbol{x}') \stackrel{\text{def}}{=} \sum_{T \in \Omega_{\text{AND}}} \underbrace{I_T^{\text{AND}} \cdot \mathbb{1}\left(\begin{smallmatrix} \boldsymbol{x}' \text{ triggers AND} \\ \text{relation between } T \end{smallmatrix}\right)}_{\text{an AND interaction}} + \sum_{T \in \Omega_{\text{OR}}} \underbrace{I_T^{\text{OR}} \cdot \mathbb{1}\left(\begin{smallmatrix} \boldsymbol{x}' \text{ triggers OR} \\ \text{relation between } T \end{smallmatrix}\right)}_{\text{an OR interaction}} + b, \tag{3}$$

where *the trigger function* $\mathbb{1}\left(\begin{smallmatrix} \boldsymbol{x}' \text{ triggers AND} \\ \text{relation between } T \end{smallmatrix}\right) \in \{0, 1\}$ *represents an AND interaction* between input variables in the set $T \subseteq N$. It returns 1 if all variables in $T$ are present (not masked[2]) in $\boldsymbol{x}'$; otherwise, it returns 0. *The trigger function* $\mathbb{1}\left(\begin{smallmatrix} \boldsymbol{x}' \text{ triggers OR} \\ \text{relation between } T \end{smallmatrix}\right) \in \{0, 1\}$ *represents an OR interaction* between input variables in the set $T \subseteq N$. It returns 1 whenever any variable in $T$ appears (not masked[2]) in $\boldsymbol{x}'$; otherwise, it returns 0. $I_T^{\text{OR}}$ and $I_T^{\text{AND}}$ are the scalar weights. $\Omega^{\text{AND}}$ and $\Omega^{\text{OR}}$ represent the set of AND interactions and the set of OR interactions extracted by the DNN from the input sample $\boldsymbol{x}'$. $b$ is a scalar bias.

**First, the fidelity requirement is satisfied by the following *universal matching property*** of interactions in Theorem 1. *I.e.*, the logical model can always accurately match the DNN's outputs $v(\boldsymbol{x}_S)$, no matter how we randomly augment the input sample $\boldsymbol{x}$ by enumerating all its $2^n$ masked states. The sample set $\Psi = \{\boldsymbol{x}_S | S \subseteq N\}$ is thus implemented as $2^n$ masked states of the input sample $\boldsymbol{x}$. $\boldsymbol{x}_S$ represents a masked input sample only containing input variables in $S$. All other variables in $N \setminus S$ are masked[2].

**Theorem 1 (Universal matching property**, proven in (Chen et al., 2024) and Appendix G**).** *Given a DNN $v$ and an input sample $\boldsymbol{x}$, let the scalar weights $I_T^{AND}$ and $I_T^{OR}$ in the logical model be set as $\forall T \subseteq N, I_T^{AND} = \sum_{L \subseteq T} (-1)^{|T| - |L|} u_L^{AND}$ and $I_T^{OR} = -\sum_{L \subseteq T} (-1)^{|T| - |L|} u_{N \setminus L}^{OR}$, where $\forall L \subseteq N$, $u_L^{AND} = 0.5 \cdot v(\boldsymbol{x}_L) + \gamma_L$ and $u_L^{OR} = 0.5 \cdot v(\boldsymbol{x}_L) - \gamma_L$. $\{\gamma_L\}$ is a set of learnable parameters. Let $b$ be set as $b = v(\boldsymbol{x}_\emptyset)$ and $\Psi$ be set as $\Psi = \{\boldsymbol{x}_S | S \subseteq N\}$. Then, we always have,*

$$\forall \boldsymbol{x}' \in \Psi, \quad h(\boldsymbol{x}') = v(\boldsymbol{x}'). \tag{4}$$

In order to compute interaction effects $I_T^{\text{AND}}$ and $I_T^{\text{OR}}$, we follow (Li & Zhang, 2023a) to learn sparse interactions by optimizing the parameters set $\{\gamma_L\}$ with a LASSO-like loss $\min_{\{\gamma_L\}} \sum_T \left( |I_T^{\text{AND}}| + |I_T^{\text{OR}}| \right)$. *Please see Appendix E for detailed pseudocode.*

**Second, the conciseness requirement is satisfied by the sparsity property.** Another issue with interactions is the conciseness of explanation. To this end, Ren et al. (2024a) have proven that, under three common conditions[3], a well-trained DNN usually encodes only $\mathcal{O}(n^\kappa / \tau) \ll 2^n$ salient

---

[2]Masking an input variable in $S$ is conducted by replacing this variable with a baseline value. The baseline value is usually set to the average value of this input variable over different samples (Dabkowski & Gal, 2017).

[3]The interaction sparsity arises naturally under three common conditions that characterize the smooth inference behavior of a DNN when evaluated on randomly masked inputs. Please see Appendix F for details.

interactions, where the absolute effects of these interactions exceed the threshold $\tau$. All other interactions have negligible effects. These salient interactions are sparse, because $\kappa \in [0.9, 1.2]$. Thus, given all potential interaction effects $\{I_T^{\mathrm{AND}}, I_T^{\mathrm{OR}} : T \subseteq N\}$ in the logical model computed in Theorem 1, we can only use salient interactions to construct a much simpler logical model by setting $\Omega_{\mathrm{sparse}}^{\mathrm{AND}} = \{T \subseteq N : |I_T^{\mathrm{AND}}| > \tau\}$ and $\Omega_{\mathrm{sparse}}^{\mathrm{OR}} = \{T \subseteq N : |I_T^{\mathrm{OR}}| > \tau\}$ to explain the DNN.

**Order of interactions.** The order of an interaction $T$ reflects the complexity of the interaction, and is defined as the number of input variables in $T$, *i.e.,* $\mathrm{order}(T) \stackrel{\mathrm{def}}{=} |T|$.

## 2.2 Towards confusing samples with non-generalizable interactions

According to Theorem 1, the AND-OR interactions derived from a DNN faithfully explain its underlying primitive inference patterns. This has motivated many researchers to define and use the generalizability of interactions to explain the generalization power of DNNs. Such studies are collectively referred to as ***symbolic generalization***, which have attracted increasing attention. For example, (1) Zhou et al. (2024) found that high-order interactions exhibited weaker generalizability than low-order ones. (2) Ren et al. (2024b) found that high-order interactions emerged primarily when the DNN kept being trained after overfitting. (3) Ren et al. (2023b) found that high-order interactions were also more difficult for Bayesian neural networks to encode than low-order ones.

However, the above studies have mainly focused on characterizing the generalizability of interactions, while **the root cause of why certain samples lead DNNs to encode non-generalizable interactions, whereas others mainly induce generalizable ones, remains unexplored and has long hampered the future development of symbolic generalization**.

Therefore, in this study, we present a counterintuitive conclusion to the above question. *I.e.,* we find that **the randomness of parameter initialization in the low layers plays a far more dominant role in determining the learning of non-generalizable interactions, which challenges the conventional belief that training samples determine the learning of non-generalizable features.**

**Quantifying the generalization power of interactions.** In order to validate our conclusion, let us first quantify the generalizability of interactions. Following (Zhou et al., 2024), an interaction extracted from a training sample is considered generalizable if it is effectively used by the DNN to classify testing samples.

However, this definition does not enable us to efficiently quantify the generalization power of a specific interaction. Thus, we apply a more efficient metric for generalizability of interactions, proposed by (He et al., 2025). Specifically, we train another DNN $v^{\mathrm{base}}$ using only the testing samples, which we refer to as the *baseline DNN*. For each salient AND interaction $S$ such that $|I_S^{\mathrm{AND}}| > \tau$, if this interaction is also encoded by the baseline DNN as a salient interaction (*i.e.,* $|I_{S,v^{\mathrm{base}}}^{\mathrm{AND}}| > \tau$), and makes similar effects on the classification (*i.e.,* $I_{S,v^{\mathrm{base}}}^{\mathrm{AND}} \cdot I_S^{\mathrm{AND}} > 0$), then it can generalize to testing samples. This is because all interactions in the *baseline DNN* are learned purely from testing data. Therefore, we can use the following binary metric $\mathcal{G}_S^{\mathrm{type}} \in \{0, 1\}$ to identify the generalizability of an AND interaction (or an OR interaction).

$$\mathcal{G}_S^{\mathrm{type}} \stackrel{\mathrm{def}}{=} \mathbb{1}(|I_{S,v^{\mathrm{base}}}^{\mathrm{type}}| > \tau) \cdot \mathbb{1}(I_{S,v^{\mathrm{base}}}^{\mathrm{type}} \cdot I_S^{\mathrm{type}} > 0), \quad \mathrm{type} \in \{\mathrm{AND}, \mathrm{OR}\}, \tag{5}$$

where $\mathbb{1}(\cdot) \in \{0, 1\}$ is an indicator function that returns 1 if the condition is satisfied, and 0 otherwise.

Then, we quantify the generalization power of interactions over different orders using the following four metrics: (1) the strength of all positive interactions of the $k$-th order $\mathbf{I}_{\mathrm{pos}}^{(k)}$, (2) the strength of all negative interactions of the $k$-th order $\mathbf{I}_{\mathrm{neg}}^{(k)}$, (3) the strength of all generalizable positive interactions $\mathbf{J}_{\mathrm{pos}}^{(k)}$, and (4) the strength of all generalizable negative interactions $\mathbf{J}_{\mathrm{neg}}^{(k)}$.

$$\mathbf{I}_{\mathrm{pos}}^{(k)} = \sum_{\mathrm{type} \in \{\mathrm{AND, OR}\}} \sum_{S \in \Omega_{\mathrm{sparse}}^{\mathrm{type}} : |S|=k} \max(I_S^{\mathrm{type}}, 0), \; \mathbf{J}_{\mathrm{pos}}^{(k)} = \sum_{\mathrm{type} \in \{\mathrm{AND, OR}\}} \sum_{S \in \Omega_{\mathrm{sparse}}^{\mathrm{type}} : |S|=k} \max(I_S^{\mathrm{type}} \cdot \mathcal{G}_S^{\mathrm{type}}, 0), \tag{6}$$

$$\mathbf{I}_{\mathrm{neg}}^{(k)} = \sum_{\mathrm{type} \in \{\mathrm{AND, OR}\}} \sum_{S \in \Omega_{\mathrm{sparse}}^{\mathrm{type}} : |S|=k} \min(I_S^{\mathrm{type}}, 0), \; \mathbf{J}_{\mathrm{neg}}^{(k)} = \sum_{\mathrm{type} \in \{\mathrm{AND, OR}\}} \sum_{S \in \Omega_{\mathrm{sparse}}^{\mathrm{type}} : |S|=k} \min(I_S^{\mathrm{type}} \cdot \mathcal{G}_S^{\mathrm{type}}, 0). \tag{7}$$

**Non-generalizable interactions are mainly encoded to represent confusing samples.** Based on the above metrics, we conducted two experiments and found that (1) non-generalizable interactions emerged only when the DNN kept being trained after overfitting, and (2) these non-generalizable

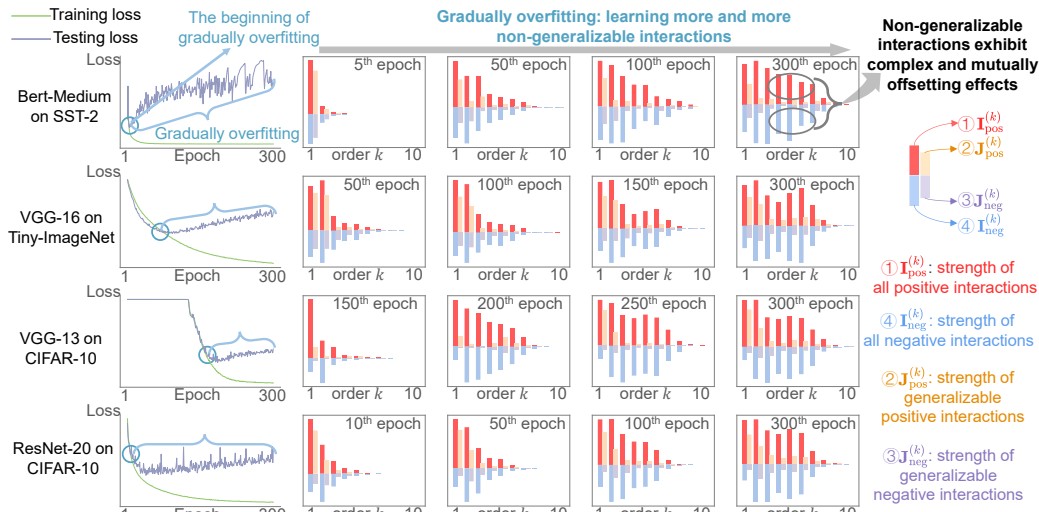

Figure 2: Changes in the distribution of interactions over different orders during the overfitting phase. The distribution of interactions is extracted from different samples. Complex and mutually offsetting interactions, which are unlikely to generalize, gradually emerge in the overfitting phase.

interactions tended to occur in only a small subset of samples rather than across all samples. We therefore defined such samples as *confusing samples*. Therefore, we can consider that **the emergence of confusing samples constitutes the root cause of a DNN's overfitting.**

• *Experiment 1: emergence of non-generalizable interactions when the DNN was overfitted.* We conducted experiments to track the strength of all interactions and the strength of generalizable interactions throughout the training process. Specifically, we trained VGG-13 (Simonyan, 2014) and ResNet-20 (He et al., 2016) on the CIFAR-10 dataset (Krizhevsky et al., 2009), VGG-16 (Simonyan, 2014) on the Tiny-ImageNet dataset (mnmoustafa, 2017), and the BERT-Medium model (Devlin, 2018) on the SST-2 dataset (Socher et al., 2013). We followed experimental settings in (Ren et al., 2024b) to select the ten patches in each image as input variables for vision tasks. We treated the embedding of each word as an input variable for NLP tasks. Please see Appendix I.3 for details. We followed (Ren et al., 2024b) to set the threshold as $\tau = 0.02 \cdot \mathbb{E}_{\boldsymbol{x}}[|v(\boldsymbol{x}) - v(\boldsymbol{x}_\emptyset)|]$.

Figure 2 shows the strength of all interactions over different orders (*i.e.,* $\mathbf{I}_{\text{pos}}^{(k)}$ and $\mathbf{I}_{\text{neg}}^{(k)}$), and the strength of generalizable interactions over different orders (*i.e.,* $\mathbf{J}_{\text{pos}}^{(k)}$ and $\mathbf{J}_{\text{neg}}^{(k)}$). Then, we observed that non-generalizable interactions were mainly learned when the overfitting problem gradually emerged (when the gap between the training loss and the testing loss significantly increased). **The emerged non-generalization interactions usually exhibited high orders and mutually offsetting effects.** Therefore, the encoding of mutually offsetting and high-order interactions could be considered a direct cause of the DNN's gradual transition into the overfitting phase.

• *Experiment 2: non-generalizable interactions only appear in confusing samples.* We find that DNNs tend to encode non-generalizable interactions only on a small set of confusing samples. For most other samples, the encoded interactions are of relatively low complexity. This observation will be further illustrated in later experiments (see Figure 5 for details).

**Identifying confusing samples to analyze the generalization power of a DNN.** We find that non-generalizable interactions in confusing samples typically exhibit two key characteristics: (1) they are of high order, and (2) their interaction effects tend to mutually offset one another (see Figure 2). Therefore, we propose to use the average order of interactions $\eta^{\text{avg}}$ extracted from each sample $\boldsymbol{x}$ as an approximate yet efficient metric to identify confusing samples. $\eta^{\text{avg}}$ is defined as follows:

$$\eta^{\text{avg}} \stackrel{\text{def}}{=} \sum\nolimits_{k=1}^{n} (k \cdot \mathbf{I}_{\text{pos}}^{(k)} + k \cdot |\mathbf{I}_{\text{neg}}^{(k)}|) / \sum\nolimits_{k=1}^{n} (\mathbf{I}_{\text{pos}}^{(k)} + |\mathbf{I}_{\text{neg}}^{(k)}|). \tag{8}$$

*Despite the simplifying assumptions underlying the metric $\eta$, the faithfulness of $\eta$ has been verified in experiments in Appendix C. Specifically, Figure 9 confirms its reliability, and Figure 8 further demonstrates that interactions extracted from confusing samples (with large $\eta^{\text{avg}}$ values) exhibit substantially lower generalization power compared to those in normal samples.*

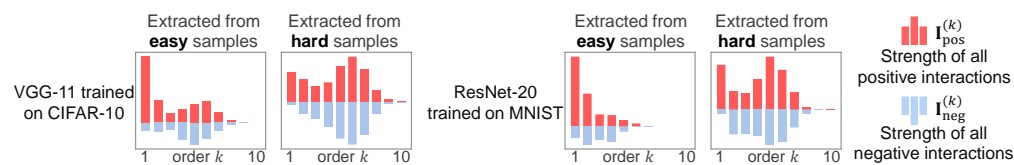

Figure 3: Distribution of interactions over different orders. We compare the distribution of interactions extracted from all hard samples and those extracted from all easy samples.

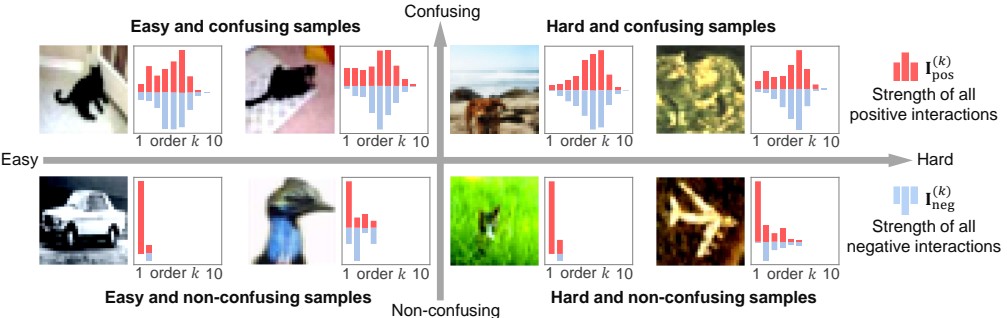

Figure 4: Comparing interaction distributions between (1) easy and confusing samples, (2) hard and confusing samples, (3) easy and non-confusing samples, and (4) hard and non-confusing samples.

### 2.3 CONFUSING SAMPLES VS. HARD SAMPLES

Hard samples are usually defined as samples with large loss (Lin, 2017), and all other normal samples are termed easy samples. We conducted experiments to explore the relationship between hard samples[4] and confusing samples. Specifically, we trained VGG-11 on the CIFAR-10 dataset and trained ResNet-20 on the MNIST dataset. Following the setting in Section 2.2, we visualized the interaction distributions of hard samples[4] and easy samples[4] in Figure 3. Additionally, we respectively presented the examples of easy confusing samples, easy non-confusing samples, hard confusing samples, and hard non-confusing samples in Figure 4.

**First, for hard samples,** we found that they could be divided into two types from the perspective of interactions. (1) As Figure 3 shows, most hard samples contained a large number of high-order and mutually offsetting interactions, which diminished the DNN's classification confidence. This indicates that most hard samples can also be considered confusing samples. (2) As Figure 4 shows, the second type of hard samples only contained a few interactions. It was the small number of interactions, not the mutually offsetting of interactions, that diminished the DNN's classification confidence. Therefore, only the first type of hard samples could be identified as confusing samples.

**Second, for easy samples,** as shown in Figure 4, they could also be divided into two types. (1) Most easy samples only contained low-order interactions, which indicated that they were not confusing samples. This was also verified by the average interaction distribution extracted from easy samples in Figure 3. (2) The second type of easy samples contained a large number of interactions, including both mutually offsetting interactions and non-offsetting low-order interactions. According to Equation (3), such numerous interactions may also boost the DNN's classification confidence.

*Traditionally, hard samples are believed to be the primary factor that pushes a DNN towards overfitting. However, our experiments in Section 2.2 show that it is confusing samples, rather than hard samples, that played a distinctive role for the overfitting of a DNN.*

## 3 EXPLORING THE DECISIVE FACTOR FOR CONFUSING SAMPLES

As one of the core challenges in the field of symbolic generalization (all papers are surveyed in Appendix A), it is still unclear which factor determines the learning of non-generalizable interactions (*i.e.,* the composition of confusing samples). Therefore, in this study, we conduct experiments and

---

[4]We computed the average classification loss of each sample across different epochs during the training process. We selected the 30 samples with the highest average loss as hard samples, and randomly selected the other 30 samples from the remaining samples as easy samples. Please see Appendix I.6 for details.

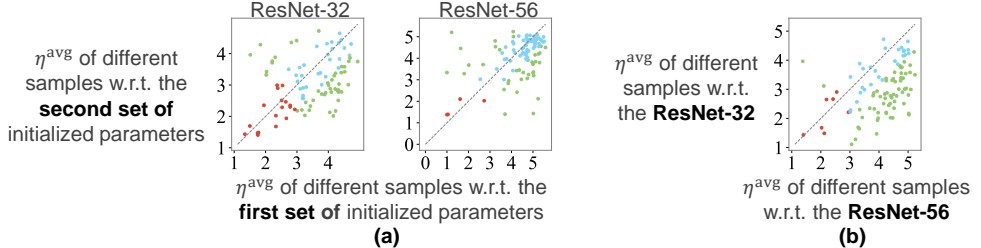

Figure 5: Comparing the composition of confusing samples of two DNNs, which are trained on the same dataset. (a) The DNNs with the **same** architecture but **different** initialized parameters had different confusing samples. (b) The DNNs with **different** architectures (of course, different low-layer parameters) had different confusing samples.

find that *it is the randomness of initialized parameters in low layers that determines the composition of confusing samples of a DNN. In comparison, other factors, such as the network's architecture and the parameters in the high layers, have much less impact on the composition of confusing samples.*

### 3.1 RANDOMNESS OF CONFUSING SAMPLES

We find a counter-intuitive phenomenon, *i.e.,* different DNNs with similar classification performance usually have fully different sets of confusing samples. This finding seems to conflict with another closely related topic, *i.e., mining hard samples*, which considers that the composition of hard samples is an intrinsic property of data distribution in a high-dimensional space. This idea has been widely used for data augmentation (Shrivastava et al., 2016; Smirnov et al., 2018; Peng et al., 2018).

However, the following phenomenon of the different DNNs having different sets of confusing samples challenges the above well-known common sense. Later, this phenomenon is found to be attributed to the randomness of initialized parameters in low layers of the DNN in Section 3.2.

**Phenomenon 1.** *DNNs with similar classification accuracies, even those with the same architecture, usually had completely different sets of confusing samples.*

We use the scatter diagram in Figure 5 to identify whether two DNNs have similar sets of confusing samples. Each point in the figure represents a sample. The horizontal axis shows a sample's average interaction order $\eta^{\mathrm{avg}}$ extracted from a DNN, and the vertical axis shows its average interaction order $\eta^{\mathrm{avg}}$ extracted from the other DNN. If the two DNNs have similar sets of confusing samples, then most points (samples) will appear near the main diagonal of the figure. Otherwise, if many confusing samples for a DNN are not confusing samples for the other DNN, then these samples would deviate from the main diagonal of the figure.

**Experiments.** We conducted experiments on ResNet-32 and ResNet-56 trained on the CIFAR-10 dataset. Figure 5(a) shows that, if parameters in two DNNs were initialized differently, then the two DNNs usually had fully different sets of confusing samples, *i.e.,* most samples deviated from the main diagonal of the figure. Figure 5(b) shows that DNNs with different architectures (thereby, having different low-layer parameters) had completely different sets of confusing samples.

**Challenging the traditional view of the samples' difficulty.** The above experiments challenge the common belief that the difficulty of samples in a dataset is an intrinsic property of the data itself, although the data simplicity still cannot be ignored, either. In other words, previous studies (Forouzesh & Thiran, 2024) usually believe that different models have similar sets of hard samples. Although confusing samples are not fully equivalent to the hard samples, our finding suggests that the simplicity of a sample is not the only factor that determines a confusing sample, especially for the DNN. Instead, later experiments in Section 3.2 will show that it is the randomness of initialized parameters in low layers that determines the composition of confusing samples.

### 3.2 IMPACT OF PARAMETERS IN LOW LAYERS

In this subsection, we conducted experiments to analyze the impact of a DNN's low-layer parameters on the composition of confusing samples.

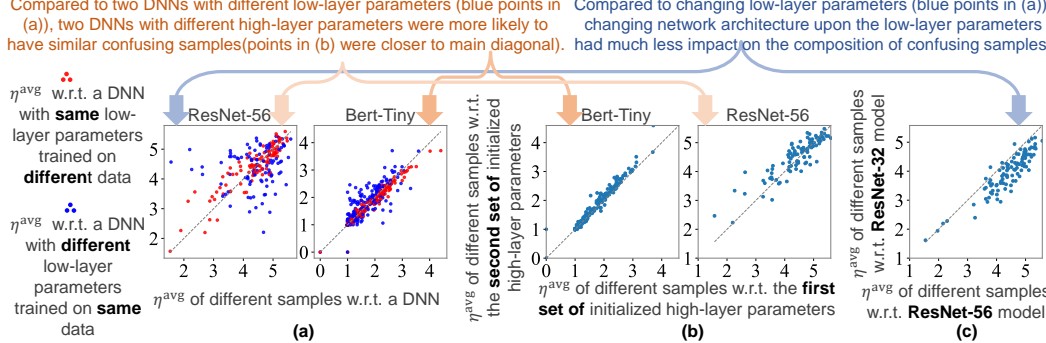

Figure 6: (a) Red points compare composition of confusing samples of two DNNs with the **same** low-layer parameters, which are trained on **different** datasets. Blue points compare the composition of confusing samples of two DNNs with the **different** low-layer parameters, which are trained on the **same** datasets. (b) Comparing the composition of confusing samples of two DNNs with the **same** low-layer parameters but **different** high-layer parameters, which are trained on the **same** datasets. (c) Comparing the composition of confusing samples of two DNNs with the **same** low-layer parameters but **different** architectures, which are trained on the **same** datasets.

**Impact of low-layer parameters.** The first experiment compared the confusing samples extracted from two DNNs with the same architecture but different parameters in the low layers. Both DNNs were initialized to have the same parameters in the high layers. As the only difference between them, we set parameters in low layers to have fully different sets of values[5]. Then, the two DNNs were trained on the same dataset to ensure a fair comparison. We trained the ResNet-56 models on the CIFAR-10 dataset and trained the Bert-Tiny models on the SST-2 dataset. Specifically, we empirically considered the first 9 convolutional layers of ResNet-56 as the low layers, and considered all the other 47 layers as the high layers. For the Bert-Tiny model, we considered the first transformer block as the low layers, and considered all layers after the first transformer block as the high layers.

Blue points in Figure 6(a) compare two sets of confusing samples[6] extracted from two DNNs with different low-layer parameters. *We found that DNNs with different low-layer parameters usually had completely different sets of confusing samples, demonstrating the impact of low-layer parameters.*

**Comparison with impact of training samples.** The second experiment illustrated the impact of training samples on the composition of confusing samples[6], which allows us to compare the impact of training samples with the impact of low-layer parameters in Figure 6(a). For implementation, we set two DNNs, which had the same architecture, to have the same low-layer parameters[7] and the same initialized high-layer parameters. The two DNNs were trained on different datasets. Specifically, we sampled two disjoint training sets from the CIFAR-10 training set for two ResNet-56 models. Similarly, we randomly sampled two disjoint sets of training samples from the SST-2 training set for two Bert-Tiny models. Please see Appendix I.6 for details.

Red points in Figure 6(a) compare two sets of confusing samples[6] extracted from the two DNNs. *We found that, compared to changing low-layer parameters in the first experiment (see blue points in Figure 6(a)), changing training samples in the second experiment (red points) was less likely to jumble up the composition of confusing samples.*

**Parameters in how many layers are sufficient to determine confusing samples?** We further conducted experiments to explore controlling parameters in how many low layers were sufficient to determine the composition of confusing samples. Specifically, let two target DNNs with the same architecture have the same low-layer parameters, *i.e.,* copying their parameters in the first $L$ layers from the same trained DNN. Then, we trained the upper layers of the two DNNs on two different sets of training samples.

---

[5]To eliminate confounding factors, we used low-layer parameters of another two DNNs, which had the same architecture and had been well trained, to replace low-layer parameters of the current two DNNs, respectively.

[6]The identification of confusing samples was conducted on testing samples, not training samples, in all experiments for a fair comparison.

[7]To eliminate confounding factors, we used low-layer parameters of another well-trained DNN to replace low-layer parameters of the current two DNNs, to let the two DNNS have the same low-layer parameters.

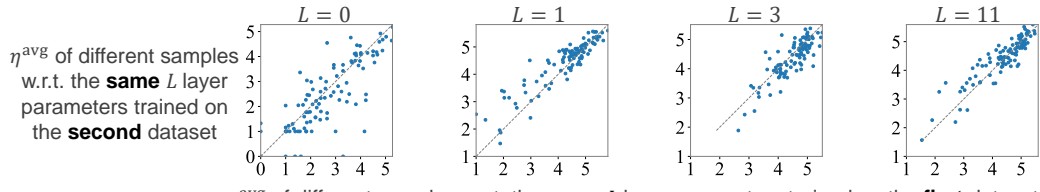

$\eta^{\mathrm{avg}}$ of different samples w.r.t. the **same** $L$ layer parameters trained on the **first** dataset

Figure 7: Comparing the composition of confusing samples of two DNNs, when the two DNNs had the same parameters of the first $L$ layers. The two DNNs were trained on different datasets.

We trained four different pairs of DNNs. The four pairs of DNNs shared parameters of the first 11 layers, the first 3 layers, the first 1 layer, and 0 layer (no parameter sharing), respectively. Each pair of DNNs was trained on two disjoint sets of training samples sampled from the CIFAR-10 training set (see Appendix I.6). Figure 7 shows that *sharing parameters in the first layer had already been enough to let two DNNs have similar sets of confusing samples[6], even though the two DNNs were trained on different training samples.*

### 3.3 Impact of high-layer parameters and network architectures

**Impact of high-layer parameters.** We conducted an experiment to analyze the impact of parameters in high layers of a DNN. Specifically, we constructed two DNNs with the same architecture, but set them to have fully different initial parameters in the high layers. In comparison, the two DNNs were set with the same low-layer parameters[7].

We trained such a pair of ResNet-56 models on the CIFAR-10 dataset, and trained a pair of Bert-Tiny models on the SST-2 dataset. We followed the experimental setting of high/low layers in Section 3.2. Figure 6(b) shows that *DNNs with different high-layer parameters still had similar sets of confusing samples[6], which suggested that parameters in high layers had relatively little impact.*

**Impact of network architectures.** We conducted an experiment to analyze the impact of network architectures on the composition of confusing samples. Specifically, we followed experimental settings in Section 3.2 to set two DNNs to have the same parameters in low layers, but the two DNNs were constructed to have fully different architectures beyond the low layers. We trained ResNet-56 and ResNet-32 on the CIFAR-10 dataset, and empirically set the first 9 layers of the two DNNs to have the same parameters[7]. Figure 6(c) shows that *DNNs with different architectures also had a similar set of confusing samples[6], indicating that network architectures had a weak impact.*

## 4 Conclusions and discussion

In this paper, we have demonstrated that the learning of complex and mutually offsetting interactions within a small set of confusing samples explains the internal mechanism of a DNN's non-generalizable representations, addressing the central challenge in the field of symbolic generalization. Moreover, we have discovered that DNNs often have fully different sets of confusing samples. It is the randomness of initialized parameters in low layers that determines the composition of confusing samples of the DNN. In comparison, other factors, such as high-layer parameters and network architecture, have much less impact on the composition of confusing samples.

**Discussion.** Our findings extend the lottery ticket hypothesis (Frankle & Carbin, 2018). The lottery ticket hypothesis suggests that the representation of a DNN is dominated by a small set of randomly initialized parameters, which are termed *winning tickets*. To this end, our experiments further showed that it was the randomness of initialized parameters in low layers that determined the composition of confusing samples of the DNN. In comparison, other factors, such as high-layer parameters and network architecture, had much less impact.

More importantly, the follow-up studies presented in Appendix D indicate that, the removal of non-generalizable interactions (*i.e.,* reducing the prevalence of confusing samples) can lead to a modest improvement in the testing performance of DNNs. Therefore, identifying the decisive factor that determines a DNN to learn confusing samples provides a new perspective for understanding DNNs' training process.

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

## A  RELATED WORK

The explainability of deep neural networks (DNNs) has received increasing attention in recent years. However, there has long been a pessimistic view regarding the possibility of faithfully explaining DNNs' inference patterns (Dziugaite & Roy, 2017; Foret et al., 2020; Neyshabur et al., 2015). Fortunately, recent advancements in interaction-based explanations, as surveyed by (Ren et al., 2024a), have made the first attempt to tackle the mathematical feasibility of explaining a DNN's inference logics using a small number of inference patterns. Specifically, (1) Ren et al. (2023a) discovered and Ren et al. (2024a) proved that there exists an AND-OR logical model, which contains only a small number of interactions, can faithfully explain the inference logics of DNNs, regardless of how the input samples are masked. (2) Zhou et al. (2024) used the complexity of interactions to explain the generalization power of DNNs. (3) Deng et al. (2024) demonstrated that fourteen attribution methods can all be explained as a reallocation of interaction effects.

Furthermore, symbolic generalization theory also provides a new perspective to explain the generalization power of a DNN. Li & Zhang (2023b) have discovered that many interactions encoded by a DNN can generalize across different samples. In particular, Zhou et al. (2024) have found that simple interactions have higher generalization power than complex interactions. Moreover, Zhang et al. (2024) discovered and Ren et al. (2024b) proved a two-phase dynamics of learning interactions during the training process of a DNN, and they found that numerous complex interactions emerge in the overfitting phase of the DNN.

In this way, compared to previous studies, this paper provides further insights into the underlying factors contributing to the overfitting of DNNs and identifies the key factor that determines the composition of confusing samples in DNNs. The lottery ticket hypothesis(Frankle & Carbin, 2018) suggests that a DNN's representation is largely influenced by a small subset of randomly initialized parameters, known as winning tickets. Building on this, our experiments showed that the low-layer parameters of a DNN are the primary determinant of the composition of confusing samples. In contrast, other factors, such as high-layer parameters and network architecture, have significantly less impact.

## B  THE USE OF LARGE LANGUAGE MODELS (LLMs)

In this work, a large language model (LLM) was used exclusively to enhance the clarity and readability of the manuscript by refining its language and phrasing. All research activities were carried out independently by the authors, and the LLM made no contribution to the research ideation or substantive content.

## C  FAITHFULNESS OF THE METRIC TO IDENTIFY THE CONFUSING SAMPLES

To validate the faithfulness of the proposed metric $\eta^{\mathrm{avg}}$, we conducted an experiment to evaluate the generalization power of interactions in samples with different $\eta^{\mathrm{avg}}$ values.

Specifically, we randomly selected two groups of samples from the testing set of CIFAR-10: one group with high $\eta^{\mathrm{avg}}$ values and the other with low $\eta^{\mathrm{avg}}$ values. For each sample, we computed the ratio of generalized interactions to total interactions, denoted as:

$$\mathcal{H} = \sum_{k=1}^{10} \frac{\mathbf{J}_{\mathrm{pos}}^{(k)} + |\mathbf{J}_{\mathrm{neg}}^{(k)}|}{\mathbf{I}_{\mathrm{pos}}^{(k)} + |\mathbf{I}_{\mathrm{neg}}^{(k)}|}.$$

We trained a VGG-11 model on CIFAR-10 and extracted interactions for this analysis. Figure 8 shows that samples with large $\eta^{\mathrm{avg}}$ values exhibited significantly lower values of $\mathcal{H}$, indicating that samples with large $\eta^{\mathrm{avg}}$ values contain numerous non-generalizable interactions. This supports our interpretation that high-$\eta^{\mathrm{avg}}$ samples typically contain a large number of high-order, mutually offsetting interactions, and thus tend to be confusing samples.

We also showed the values of $\mathcal{H}_{\mathrm{low}} = \sum_{k=1}^{2} \frac{\mathbf{J}_{\mathrm{pos}}^{(k)} + |\mathbf{J}_{\mathrm{neg}}^{(k)}|}{\mathbf{I}_{\mathrm{pos}}^{(k)} + |\mathbf{I}_{\mathrm{neg}}^{(k)}|}$   and   $\mathcal{H}_{\mathrm{high}} = \sum_{k=6}^{10} \frac{\mathbf{J}_{\mathrm{pos}}^{(k)} + |\mathbf{J}_{\mathrm{neg}}^{(k)}|}{\mathbf{I}_{\mathrm{pos}}^{(k)} + |\mathbf{I}_{\mathrm{neg}}^{(k)}|}$.

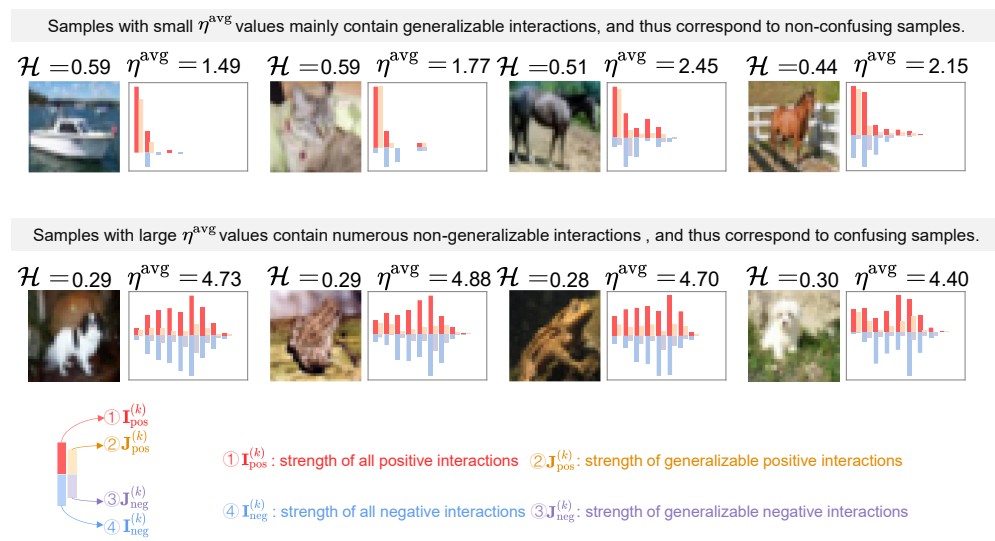

Figure 8: Faithfulness of the Metric $\eta^{\text{avg}}$ to identify the confusing samples. Sample with high $\eta^{\text{avg}}$ values usually contains a large number of high-order, mutually offsetting interactions, and thus tend to be confusing samples.

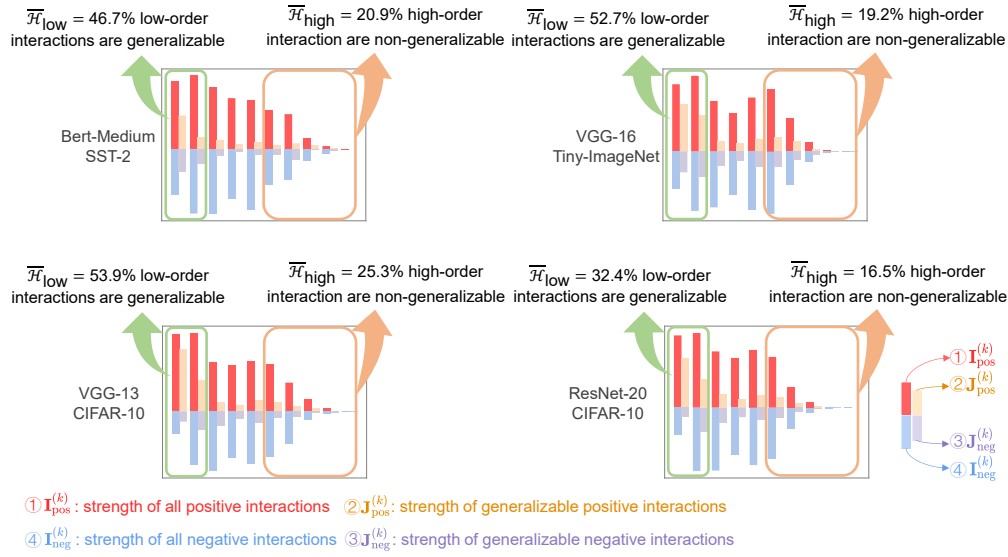

Figure 9: Faithfulness of the Metric $\eta^{\text{avg}}$ to identify the confusing samples. High-order interactions had significantly weaker generalization power than low-order ones. Samples with high $\eta^{\text{avg}}$ values contain numerous non-generalizable high-order interaction, and thus tend to be confusing samples.

We trained ResNet-20 and VGG-13 on the CIFAR-10 dataset, BERT-Medium on the SST-2 dataset, and VGG-16 on the Tiny-ImageNet dataset. Figure 9 shows the values of $\mathcal{H}_{\text{low}}$ and $\mathcal{H}_{\text{high}}$ for these different DNNs. Experimental results demonstrated that high-order interactions had significantly weaker generalization power than low-order interactions. This further supported the faithfulness of the $\eta^{\text{avg}}$ metric, as samples with high $\eta^{\text{avg}}$ values tended to contain a larger proportion of non-generalizable high-order interactions.

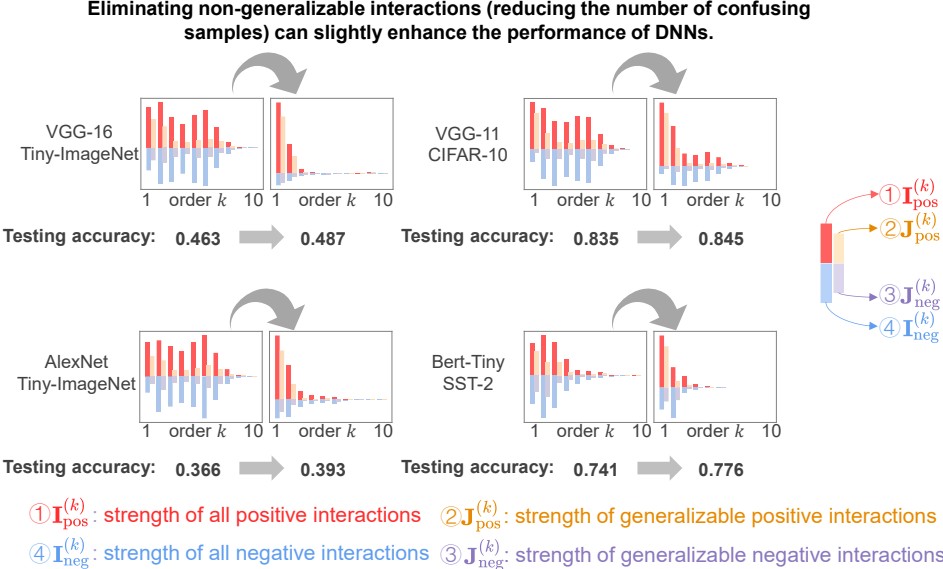

Figure 10: Changes in the distribution of interactions and the corresponding performance (testing accuracy) of the models before and after eliminating non-generalizable interactions. Eliminating non-generalizable interactions (*i.e.,* reducing the number of confusing samples) can yield moderate improvements in model performance.

# D PRACTICAL VALUE

**Improving the performance of models in practice.** In fact, we observe that eliminating non-generalizable interactions (*i.e.,* reducing the number of confusing samples) can lead to moderate improvements in model performance. During training, we explicitly removed the non-generalizable interactions encoded by the DNN (*i.e.,* reducing the number of confusing samples) and subsequently observed a measurable gain in performance. Specifically, we compared the distribution of of interactions over orders and the corresponding changes in model performance (testing accuracy) before and after eliminating non-generalizable interactions. We conducted experiments by training VGG-16 and AlexNet (Krizhevsky et al., 2012) on the Tiny-ImageNet dataset, BERT-Tiny on the SST-2 dataset, and VGG-11 on the CIFAR-10 dataset.

Figure 10 illustrates the changes in the distribution of interactions and the corresponding performance (testing accuracy) of the models before and after eliminating non-generalizable interactions. *These results demonstrated that eliminating non-generalizable interactions (*i.e., *reducing the number of confusing samples) could yield moderate improvements in model performance.*

# E DETAILS TO EXTRACT THE SPARSEST AND-OR INTERACTIONS

A method is proposed (Li & Zhang, 2023a; Chen et al., 2024) to simultaneously extract AND interactions $I_T^{\text{AND}}$ and OR interactions $I_T^{\text{OR}}$ from the network output. Given a masked sample $\boldsymbol{x}_L$, Li & Zhang (2023a) proposed to learn a decomposition $v(\boldsymbol{x}_L) = u_L^{\text{AND}} + u_L^{\text{OR}}$ towards the sparsest interactions. The component $u_L^{\text{AND}}$ was explained by AND interactions, and the component $u_L^{\text{OR}}$ was explained by OR interactions. Specifically, they decomposed $v(\boldsymbol{x}_L)$ into $u_L^{\text{AND}} = 0.5 \cdot v(\boldsymbol{x}_L) + \gamma_L$ and $u_L^{\text{OR}} = 0.5 \cdot v(\boldsymbol{x}_L) - \gamma_L$, where $\{\gamma_L : L \subseteq N\}$ is a set of learnable variables that determine the decomposition. In this way, the AND interactions and OR interactions can be computed according to Theorem 1, *i.e.,* $I_T^{\text{AND}} = \sum_{L \subseteq T}(-1)^{|T|-|L|}u_L^{\text{AND}}$, and $I_T^{\text{OR}} = -\sum_{L \subseteq T}(-1)^{|T|-|L|}v_{N \setminus L}^{\text{OR}}$.

The parameters $\{\gamma_L\}$ were learned by minimizing the following LASSO-like loss to obtain sparse interactions:

$$\min_{\{\gamma_L\}} \sum_{T \subseteq N} |I_T^{\text{AND}}| + |I_T^{\text{OR}}| \tag{9}$$

**Removing small noises.** A small noise $\delta$ in the network output may significantly affect the extracted interactions, especially for high-order interactions. Thus, Li & Zhang (2023a) proposed to learn to remove a small noise term $\delta_T$ from the computation of AND-OR interactions. Specifically, the decomposition was rewritten as $u_L^{\text{AND}} = 0.5(v(\boldsymbol{x}_L) - \delta_L) + \gamma_L$ and $u_L^{\text{OR}} = 0.5(v(\boldsymbol{x}_L) - \delta_L) + \gamma_L$. Thus, the parameters $\{\delta_L\}$ and $\{\gamma_L\}$ are simultaneously learned by minimizing the loss function in Eq. (9). The values of $\{\delta_L\}$ were constrained in $[-\zeta, \zeta]$ where $\zeta = 0.01 \cdot |v(\boldsymbol{x}) - v(\boldsymbol{x}_\emptyset)|$.

**Algorithm of extracting AND-OR interactions.** The techinical details of computing $I_T^{\text{AND}}$ and $I_T^{\text{OR}}$ is provided in the following pseudocode in Algorithm 1.

---

**Algorithm 1** Compute AND and OR Interactions and Select Salient Ones

---

**Require:** Deep neural network $v$, input sample $\boldsymbol{x} = [x_1, x_2, \ldots, x_n]^T$, set of indices $N = \{1, 2, \ldots, n\}$, small noise threshold $\zeta$, significance threshold $\tau$, convergence threshold $\epsilon$.
**Ensure:** AND interactions $I_T^{\text{AND}}$, OR interactions $I_T^{\text{OR}}$, and significant interaction sets $\Omega_{\text{sparse}}^{\text{AND}}$ and $\Omega_{\text{sparse}}^{\text{OR}}$.
 1: Initialize learnable parameters $\{\gamma_L\}$ and $\{\delta_L\}$ for all $L \subseteq N$.
 2: Compute baseline output $v(\boldsymbol{x}_\emptyset)$, where $\boldsymbol{x}_\emptyset$ is the masked sample with all variables removed.
 3: Initialize previous loss $\mathcal{L}_{\text{prev}} \leftarrow \infty$.
 4: **repeat**
 5:     **for** each subset $L \subseteq N$ **do**
 6:         Compute masked sample $\boldsymbol{x}_L$ by removing variables not in $L$.
 7:         Compute network output $v(\boldsymbol{x}_L)$.
 8:         Compute noise term $\delta_L$ constrained in $[-\zeta, \zeta]$, where $\zeta = 0.01 \cdot |v(\boldsymbol{x}) - v(\boldsymbol{x}_\emptyset)|$.
 9:         Decompose $v(\boldsymbol{x}_L)$ into AND and OR components:
10:         $u_L^{\text{AND}} \leftarrow 0.5 \cdot (v(\boldsymbol{x}_L) - \delta_L) + \gamma_L$
11:         $u_L^{\text{OR}} \leftarrow 0.5 \cdot (v(\boldsymbol{x}_L) - \delta_L) - \gamma_L$
12:     **end for**
13:     **for** each subset $T \subseteq N$ **do**
14:         Compute AND interaction $I_T^{\text{AND}}$:

$$I_T^{\text{AND}} \leftarrow \sum_{L \subseteq T} (-1)^{|T|-|L|} u_L^{\text{AND}}$$

15:         Compute OR interaction $I_T^{\text{OR}}$:

$$I_T^{\text{OR}} \leftarrow - \sum_{L \subseteq T} (-1)^{|T|-|L|} u_{N \setminus L}^{\text{OR}}$$

16:     **end for**
17:     Compute current loss $\mathcal{L} \leftarrow \sum_{T \subseteq N} \left( |I_T^{\text{AND}}| + |I_T^{\text{OR}}| \right)$.
18:     Optimize parameters $\{\gamma_L\}$ and $\{\delta_L\}$ to minimize $\mathcal{L}$.
19:     Check for convergence: $|\mathcal{L} - \mathcal{L}_{\text{prev}}| < \epsilon$.
20:     Update previous loss: $\mathcal{L}_{\text{prev}} \leftarrow \mathcal{L}$.
21: **until** convergence
22: Select significant AND interactions: $\Omega_{\text{sparse}}^{\text{AND}} \leftarrow \{T \subseteq N : |I_T^{\text{AND}}| > \tau\}$
23: Select significant OR interactions: $\Omega_{\text{sparse}}^{\text{OR}} \leftarrow \{T \subseteq N : |I_T^{\text{OR}}| > \tau\}$
24: **return** $I_T^{\text{AND}}$, $I_T^{\text{OR}}$, $\Omega_{\text{sparse}}^{\text{AND}}$, and $\Omega_{\text{sparse}}^{\text{OR}}$.

---

## F    COMMON CONDITIONS FOR SPARSE INTERACTIONS

Ren et al. (2024a) have proved three sufficient conditions for the sparsity of AND interactions.

**Condition 1.** *The DNN does not encode extremely high-order interactions:* $\forall\, T \in \{T \subseteq N \,|\, |T| \geq M + 1\}$, $I_T^{\mathrm{and}} = 0$.

Condition 1 is common because extremely high-order interactions usually represent very complex and over-fitted patterns, which are unlikely to be learned by a well-trained DNN in real scenarios.

**Condition 2.** *Let $\bar{u}^{(k)} \stackrel{\text{def}}{=} \mathbb{E}_{|S|=k}[v(\boldsymbol{x}_S) - v(\boldsymbol{x}_\emptyset)]$ denote the average classification confidence of the DNN over all masked samples $\boldsymbol{x}_S$ with $k$ unmasked input variables. This average classification confidence monotonically increases when $k$ increases:* $\forall\, k' \leq k$, $\bar{u}^{(k')} \leq \bar{u}^{(k)}$.

Condition 2 implies that a well-trained DNN is likely to have higher average classification confidence for less masked input samples.

**Condition 3.** *Given the average classification confidence $\bar{u}^{(k)}$ of samples with $k$ unmasked input variables, there is a polynomial lower bound for the average classification confidence with $k'(k' \leq k)$ unmasked input variables:* $\forall\, k' \leq k$, $\bar{u}^{(k')} \geq (\frac{k'}{k})^p\, \bar{u}^{(k)}$, where $p > 0$ is a constant.

Condition 3 suggests that the classification confidence of the DNN remains relatively stable even when presented with masked input samples. In real-world applications, the classification or detection of masked or occluded samples frequently occurs. As a result, a well-trained DNN typically develops the ability to classify such masked inputs by leveraging local information, which can be derived from the visible portions of the input. Consequently, the model should not produce a substantially reduced confidence score for masked samples.

# G   PROOF OF THEOREM 1

*Proof.* **(1) Universal matching theorem of AND interactions.**

We will prove that output component $v_S^{\mathrm{AND}}$ on all $2^n$ masked samples $\{\boldsymbol{x}_S : S \subseteq N\}$ could be universally explained by the all interactions in $S \subseteq N$, *i.e.*, $\forall \emptyset \neq S \subseteq N$, $v_S^{\mathrm{AND}} = \sum_{\emptyset \neq T \subseteq S} I_T^{\mathrm{AND}} + v(\boldsymbol{x}_\emptyset)$. In particular, we define $v_\emptyset^{\mathrm{AND}} = v(\boldsymbol{x}_\emptyset)$ (*i.e.*, we attribute output on an empty sample to AND interactions).

Specifically, the AND interaction is defined as $I_T^{\mathrm{AND}} = \sum_{L \subseteq T}(-1)^{|T|-|L|} u_L^{\mathrm{AND}}$. To compute the sum of AND interactions $\sum_{\emptyset \neq T \subseteq S} I_T^{\mathrm{AND}} = \sum_{\emptyset \neq T \subseteq S} \sum_{L \subseteq T}(-1)^{|T|-|L|} u_L^{\mathrm{AND}}$, we first exchange the order of summation of the set $L \subseteq T \subseteq S$ and the set $T \supseteq L$. That is, we compute all linear combinations of all sets $T$ containing $L$ with respect to the model outputs $u_L^{\mathrm{AND}}$ given a set of input variables $L$, *i.e.*, $\sum_{T:L \subseteq T \subseteq S}(-1)^{|T|-|L|} u_L^{\mathrm{AND}}$. Then, we compute all summations over the set $L \subseteq S$.

In this way, we can compute them separately for different cases of $L \subseteq T \subseteq S$. In the following, we consider the cases (1) $L = S = T$, and (2) $L \subseteq T \subseteq S, L \neq S$, respectively.

(1) When $L = S = T$, the linear combination of all subsets $T$ containing $L$ with respect to the model output $u_L^{\mathrm{AND}}$ is $(-1)^{|S|-|S|} u_L^{\mathrm{AND}} = u_L^{\mathrm{AND}}$.

(2) When $L \subseteq T \subseteq S, L \neq S$, the linear combination of all subsets $T$ containing $L$ with respect to the model output $u_L^{\mathrm{AND}}$ is $\sum_{T:L \subseteq T \subseteq S}(-1)^{|T|-|L|} u_L^{\mathrm{AND}}$. For all sets $T : S \supseteq T \supseteq L$, let us consider the linear combinations of all sets $T$ with number $|T|$ for the model output $u_L^{\mathrm{AND}}$, respectively. Let $m := |T| - |L|$, $(0 \leq m \leq |S| - |L|)$, then there are a total of $C_{|S|-|L|}^m$ combinations of all sets $T$ of order $|T|$. Thus, given $L$, accumulating the model outputs $u_L^{\mathrm{AND}}$ corresponding to all $T \supseteq L$, then

$$\sum_{T:L \subseteq T \subseteq S}(-1)^{|T|-|L|} u_L^{\mathrm{AND}} = u_L^{\mathrm{AND}} \cdot \underbrace{\sum_{m=0}^{|S|-|L|} C_{|S|-|L|}^m (-1)^m}_{=0} = 0.$$ Please see the complete

derivation of the following formula.

$$\sum_{\emptyset \neq T \subseteq S} I_T^{\text{AND}} = \sum_{\emptyset \neq T \subseteq S} \sum_{L \subseteq T} (-1)^{|T|-|L|} u_L^{\text{AND}}$$

$$= \sum_{L \subseteq S} \sum_{T:L \subseteq T \subseteq S} (-1)^{|T|-|L|} u_L^{\text{AND}} - v_{\emptyset}^{\text{AND}}$$

$$= \underbrace{v_S^{\text{AND}}}_{L=S} + \sum_{L \subseteq S, L \neq S} u_L^{\text{AND}} \cdot \underbrace{\sum_{m=0}^{|S|-|L|} C_{|S|-|L|}^m (-1)^m}_{=0} - v_{\emptyset}^{\text{AND}} \tag{10}$$

$$= v_S^{\text{AND}} - v_{\emptyset}^{\text{AND}} = v_S^{\text{AND}} - v(\boldsymbol{x}_{\emptyset})$$

Thus, we have $\forall \emptyset \neq S \subseteq N, v_S^{\text{AND}} = \sum_{\emptyset \neq T \subseteq S} I_T^{\text{AND}} + v(\boldsymbol{x}_{\emptyset})$.

**(2) Universal matching theorem of OR interactions.**

According to the definition of OR interactions, we will derive that $\forall S \subseteq N, v_S^{\text{OR}} = \sum_{T:T \cap S \neq \emptyset} I_T^{\text{OR}}$, where we define $v_{\emptyset}^{\text{OR}} = 0$ (recall that in Step (1), we attribute the output on empty input to AND interactions).

Specifically, the OR interaction is defined as $I_T^{\text{OR}} = -\sum_{L \subseteq T} (-1)^{|T|-|L|} v_{N \setminus L}^{\text{OR}}$. Similar to the above derivation of the universal matching theorem of AND interactions, to compute the sum of OR interactions $\sum_{T:T \cap S \neq \emptyset} I_T^{\text{OR}} = \sum_{T:T \cap S \neq \emptyset} \left[ -\sum_{L \subseteq T} (-1)^{|T|-|L|} v_{N \setminus L}^{\text{OR}} \right]$, we first exchange the order of summation of the set $L \subseteq T \subseteq N$ and the set $T : T \cap S \neq \emptyset$. That is, we compute all linear combinations of all sets $T$ containing $L$ with respect to the model outputs $v_{N \setminus L}^{\text{OR}}$ given a set of input variables $L$, i.e., $\sum_{T:T \cap S \neq \emptyset, T \supseteq L} (-1)^{|T|-|L|} v_{N \setminus L}^{\text{OR}}$. Then, we compute all summations over the set $L \subseteq N$.

In this way, we can compute them separately for different cases of $L \subseteq T \subseteq N, T \cap S \neq \emptyset$. In the following, we consider the cases (1) $L = N \setminus S$, (2) $L = N$, (3) $L \cap S \neq \emptyset, L \neq N$, and (4) $L \cap S = \emptyset, L \neq N \setminus S$, respectively.

(1) When $L = N \setminus S$, the linear combination of all subsets $T$ containing $L$ with respect to the model output $v_{N \setminus L}^{\text{OR}}$ is $\sum_{T:T \cap S \neq \emptyset, T \supseteq L} (-1)^{|T|-|L|} v_{N \setminus L}^{\text{OR}} = \sum_{T:T \cap S \neq \emptyset, T \supseteq L} (-1)^{|T|-|L|} v_S^{\text{OR}}$. For all sets $T : T \supseteq L, T \cap S \neq \emptyset$ (then $T \neq N \setminus S, T \neq L$), let us consider the linear combinations of all sets $T$ with number $|T|$ for the model output $v_S^{\text{OR}}$, respectively. Let $|T'| := |T| - |L|, (1 \leq |T'| \leq |S|)$, then there are a total of $C_{|S|}^{|T'|}$ combinations of all sets $T'$ of order $|T'|$. Thus, given $L$, accumulating the model outputs $v_S^{\text{OR}}$ corresponding to all $T \supseteq L$, then $\sum_{T:T \cap S \neq \emptyset, T \supseteq L} (-1)^{|T|-|L|} v_{N \setminus L}^{\text{OR}} = v_S^{\text{OR}} \cdot$

$$\underbrace{\sum_{|T'|=1}^{|S|} C_{|S|}^{|T'|} (-1)^{|T'|}}_{=-1} = -v_S^{\text{OR}}.$$

(2) When $L = N$ (then $T = N$), the linear combination of all subsets $T$ containing $L$ with respect to the model output $v_{N \setminus L}^{\text{OR}}$ is $\sum_{T:T \cap S \neq \emptyset, T \supseteq L} (-1)^{|T|-|L|} v_{N \setminus L}^{\text{OR}} = (-1)^{|N|-|N|} v_{\emptyset}^{\text{OR}} = v_{\emptyset}^{\text{OR}}$.

(3) When $L \cap S \neq \emptyset, L \neq N$, the linear combination of all subsets $T$ containing $L$ with respect to the model output $v_{N \setminus L}^{\text{OR}}$ is $\sum_{T:T \cap S \neq \emptyset, T \supseteq L} (-1)^{|T|-|L|} v_{N \setminus L}^{\text{OR}}$. For all sets $T : T \supseteq L, T \cap S \neq \emptyset$, let us consider the linear combinations of all sets $T$ with number $|T|$ for the model output $v_S^{\text{OR}}$, respectively. Let us split $|T| - |L|$ into $|T'|$ and $|T''|$, i.e., $|T| - |L| = |T'| + |T''|$, where $T' = \{i | i \in T, i \notin L, i \in N \setminus S\}, T'' = \{i | i \in T, i \notin L, i \in S\}$ (then $0 \leq |T''| \leq |S| - |S \cap L|$) and $|T'| + |T''| + |L| = |T|$. In this way, there are a total of $C_{|S|-|S \cap L|}^{|T''|}$ combinations of all sets $T''$ of order $|T''|$. Thus, given $L$, accumulating the model outputs $v_{N \setminus L}^{\text{OR}}$ corresponding to all $T \supseteq L$, then

$$\sum_{T:T \cap S \neq \emptyset, T \supseteq L} (-1)^{|T|-|L|} v_{N \setminus L}^{\text{OR}} = v_{N \setminus L}^{\text{OR}} \cdot \sum_{T' \subseteq N \setminus S \setminus L} \underbrace{\sum_{|T''|=0}^{|S|-|S \cap L|} C_{|S|-|S \cap L|}^{|T''|} (-1)^{|T'|+|T''|}}_{=0} =$$

0.

(4) When $L \cap S = \emptyset, L \neq N \setminus S$, the linear combination of all subsets $T$ containing $L$ with respect to the model output $v_{N \setminus L}^{\text{OR}}$ is $\sum_{T:T \cap S \neq \emptyset, T \supseteq L} (-1)^{|T|-|L|} v_{N \setminus L}^{\text{OR}}$. Similarly, let us split $|T| - |L|$

into $|T'|$ and $|T''|$, *i.e.*, $|T| - |L| = |T'| + |T''|$, where $T' = \{i | i \in T, i \notin L, i \in N \setminus S\}$, $T'' = \{i | i \in T, i \in S\}$ (then $0 \leq |T''| \leq |S|$) and $|T'| + |T''| + |L| = |T|$. In this way, there are a total of $C_{|S|}^{|T''|}$ combinations of all sets $T''$ of order $|T''|$. Thus, given $L$, accumulating the model outputs $v_{N \setminus L}^{\mathrm{OR}}$ corresponding to all $T \supseteq L$, then $\sum_{T:T \cap S \neq \emptyset, T \supseteq L} (-1)^{|T|-|L|} v_{N \setminus L}^{\mathrm{OR}} = v_{N \setminus L}^{\mathrm{OR}} \cdot \sum_{T' \subseteq N \setminus S \setminus L} \underbrace{\sum_{|T''|=0}^{|S|} C_{|S|}^{|T''|} (-1)^{|T'|+|T''|}}_{=0} = 0.$

Please see the complete derivation of the following formula.

$$
\begin{aligned}
\sum_{T:T \cap S \neq \emptyset} I_T^{\mathrm{OR}} &= \sum_{T:T \cap S \neq \emptyset} \left[ -\sum_{L \subseteq T} (-1)^{|T|-|L|} v_{N \setminus L}^{\mathrm{OR}} \right] \\
&= -\sum_{L \subseteq N} \sum_{T:T \cap S \neq \emptyset, T \supseteq L} (-1)^{|T|-|L|} v_{N \setminus L}^{\mathrm{OR}} \\
&= -\left[ \sum_{|T'|=1}^{|S|} C_{|S|}^{|T'|} (-1)^{|T'|} \right] \cdot \underbrace{v_S^{\mathrm{OR}}}_{L=N \setminus S} - \underbrace{v_\emptyset^{\mathrm{OR}}}_{L=N} \\
&\quad - \sum_{L \cap S \neq \emptyset, L \neq N} \left[ \sum_{T' \subseteq N \setminus S \setminus L} \left( \sum_{|T''|=0}^{|S|-|S \cap L|} C_{|S|-|S \cap L|}^{|T''|} (-1)^{|T'|+|T''|} \right) \right] \cdot v_{N \setminus L}^{\mathrm{OR}} \\
&\quad - \sum_{L \cap S = \emptyset, L \neq N \setminus S} \left[ \sum_{T' \subseteq N \setminus S \setminus L} \left( \sum_{|T''|=0}^{|S|} C_{|S|}^{|T''|} (-1)^{|T'|+|T''|} \right) \right] \cdot v_{N \setminus L}^{\mathrm{OR}} \\
&= -(-1) \cdot v_S^{\mathrm{OR}} - v_\emptyset^{\mathrm{OR}} - \sum_{L \cap S \neq \emptyset, L \neq N} \left[ \sum_{T' \subseteq N \setminus S \setminus L} 0 \right] \cdot v_{N \setminus L}^{\mathrm{OR}} \\
&\quad - \sum_{L \cap S = \emptyset, L \neq N \setminus S} \left[ \sum_{T' \subseteq N \setminus S \setminus L} 0 \right] \cdot v_{N \setminus L}^{\mathrm{OR}} \\
&= v_S^{\mathrm{OR}} - v_\emptyset^{\mathrm{OR}} \\
&= v_S^{\mathrm{OR}}
\end{aligned}
\tag{11}
$$

**(3) Universal matching theorem of AND-OR interactions.**

With the universal matching theorem of AND interactions and the universal matching theorem of OR interactions, we can easily get $v(\boldsymbol{x}_S) = v_S^{\mathrm{AND}} + v_S^{\mathrm{OR}} = v(\boldsymbol{x}_\emptyset) + \sum_{\emptyset \neq T \subseteq S} I_T^{\mathrm{AND}} + \sum_{T:T \cap S \neq \emptyset} I_T^{\mathrm{OR}}$, thus, we obtain the universal matching theorem of AND-OR interactions.

$\square$

# H  EXAMPLES OF AND-OR LOGICAL MODELS EXPLAINING LLMS

In this section, to demonstrate that AND-OR logical models can faithfully explain large language models, we present examples showing that, given specific input prompts, AND-OR logical models can be constructed to faithfully explain both the DeepSeek-r1-distill-llama-8b (DeepSeek) (Guo et al., 2025) and Qwen2.5-7b (Qwen) (Bai et al., 2023) models. Figure 11 illustrates the AND-OR logical models used to explain DeepSeek and Qwen under different input prompts.

# I  EXPERIMENTAL DETAIL

## I.1  TRAINING SETTINGS

In this paper, we trained various DNNs for different tasks. Specifically, for the image classification task, we trained VGG-11/13 on the CIFAR-10 dataset with a learning rate of 0.01. We trained

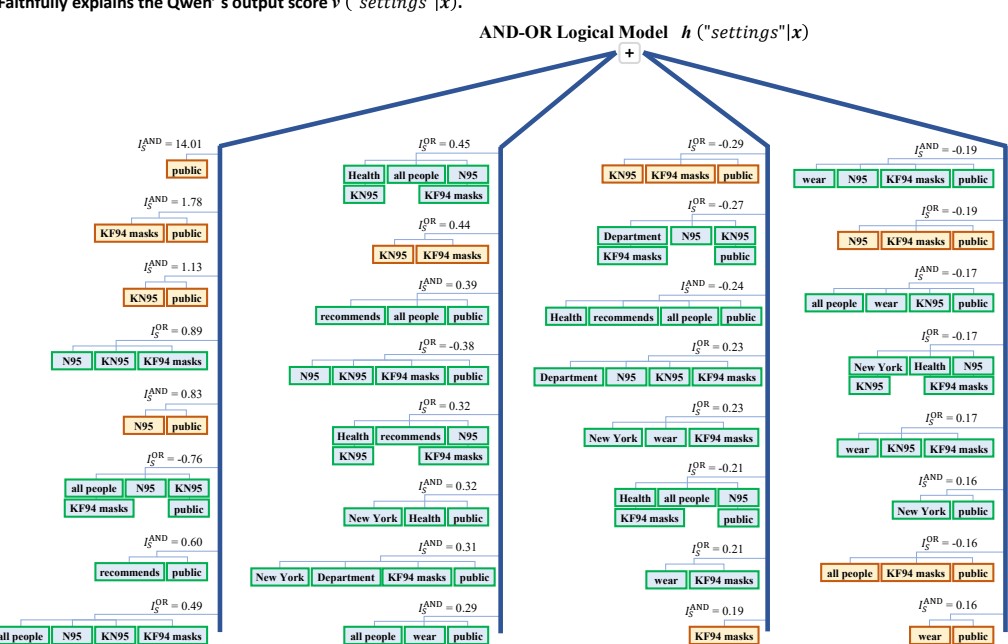

Figure 11: An example of AND-OR logical models constructed to faithfully explain the output scores of the DeepSeek model (top) and the Qwen model (bottom) on a single sample. A further example is presented on the next page.

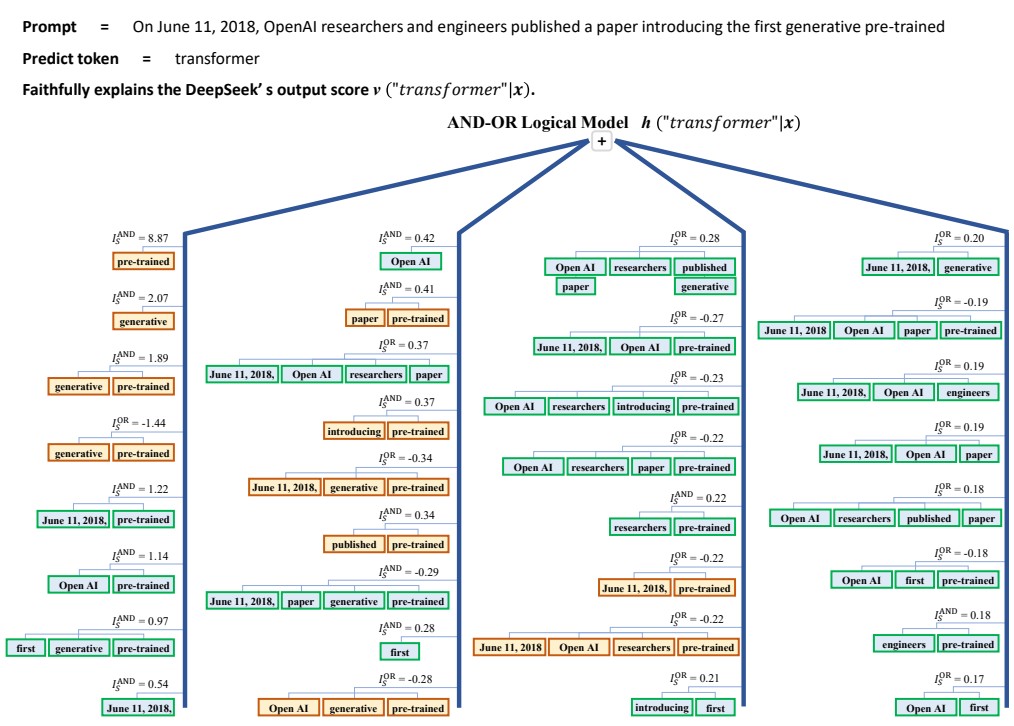

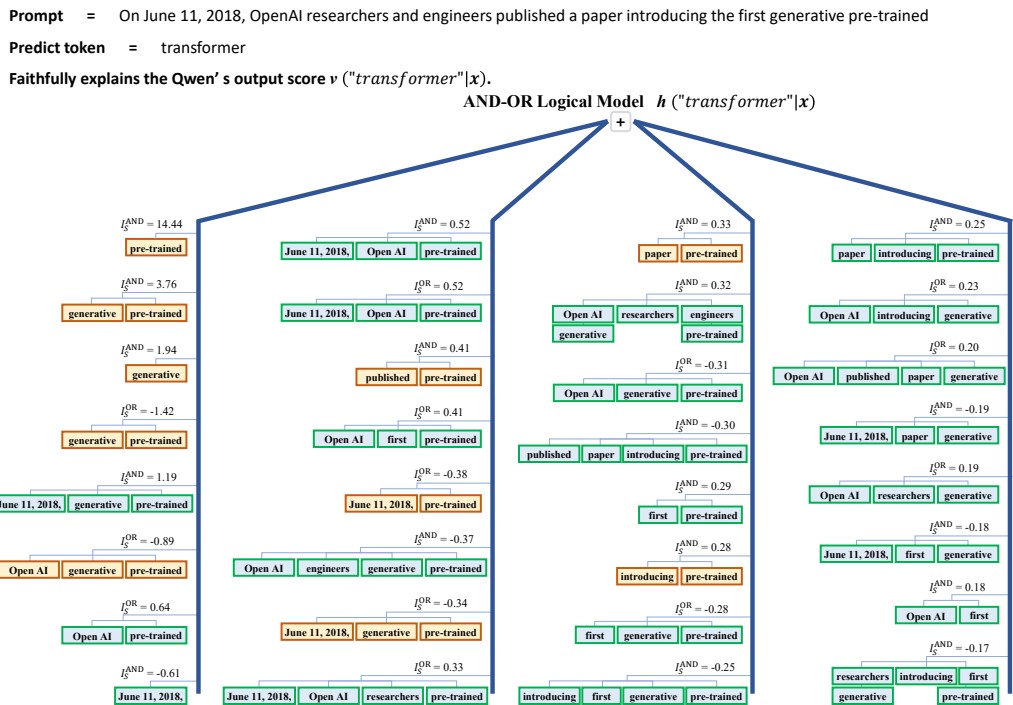

Figure 11: Another example of AND-OR logical models explaining the DeepSeek model (top) and the Qwen model (bottom) on a different sample.

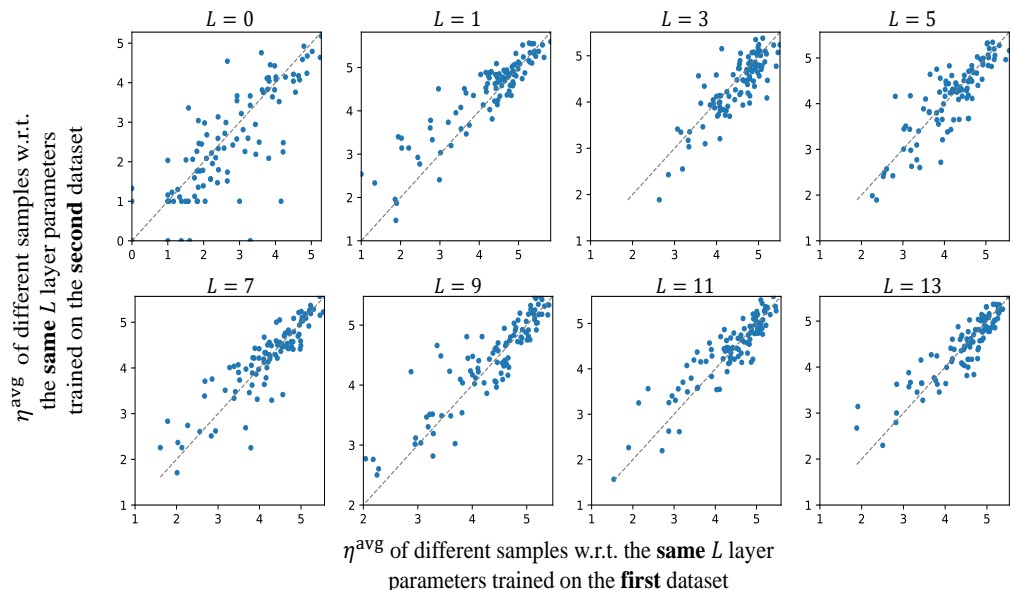

Figure 12: Comparing composition of confusing samples of two DNNs with the **same** $L$ low-layer parameters, which are trained on **different** datasets.

ResNet-20 on the CIFAR-10 dataset (MIT license) and the MNIST dataset (Creative Commons Attribution-Share Alike 3.0 license). We trained VGG-16 on the Tiny-ImageNet dataset with a learning rate of $0.01$. We trained ResNet-34/56 on the CIFAR-10 dataset with a learning rate of $0.001$. For the sentiment classification task, we trained the Bert-Tiny and Bert-Medium model on the SST-2 dataset (license unknown) with a learning rate of $0.01$. All DNNs were trained using the SGD optimizer (Robbins & Monro, 1951) with a momentum of 0.9.

For partial experiments, we adopted $\ell_\infty$-norm bounded adversarial training following the approach of (Madry, 2017). Specifically, adversarial examples were generated using a single-step Projected Gradient Descent (PGD) attack with a maximum perturbation size of $\epsilon = 4/255$, step size of $\alpha = 4/255$, and $n_{\text{step}} = 1$. Specifically, apart from the experiments in Section 2.2 (where we aimed to explore the changes in the distribution of interactions during the normal training process), all other experiments were conducted with adversarial training.

To evaluate the generalization power of interactions, we trained a DNN on the testing set as a baseline DNN to determine whether an interaction is generalizable. To ensure a fair comparison (*i.e.,* the same number of training samples for each model), we randomly split the training set of each dataset into two equal parts. One part was used as the training set and the other as the testing set. We then trained separate models on each subset and measured the generalization power of interactions accordingly.

## I.2    COMPUTER RESOURCES

All DNNs were trained on a single NVIDIA GeForce RTX 3080 Ti (with 12G GPU memory), and can be finished within 24 hours. Calculating all interactions for an input sample takes an average of 30-50 seconds.

## I.3    DETAILS ABOUT HOW TO CALCULATE INTERACTIONS FOR DIFFERENT DNNS

• **For image data in different image datasets,** since the computational cost of interactions was intolerable, we applied a sampling-based approximation method to calculate AND-OR interactions. Specifically, we considered the feature map after the low-layer as intermediate-layer features of DNNs. We uniformly split the central region of each intermediate-layer feature (*i.e.,* we did not consider the pixel on the edges of an image) into $5 \times 5$ patches and selected the 10 patches with

the highest L1 norms (*i.e.,* the brightest ones) to calculate interactions, and considered these patches as input variables for each intermediate-layer feature. We used **0** as a baseline value to mask the variables in $N \backslash T$.

• **For natural language data in SST-2 dataset,** we considered the outputs of the low-layer corresponding to input words as input features. We considered the embeddings corresponding to input features as input variables for each input sentence, and we randomly sampled 10 words, which must have a specific meaning and not be stop words, to calculate interactions. We used the average embedding over different input variables to mask the tokens in $N \backslash T$.

Specifically, we empirically considered the first 9 convolutional layers of ResNet-56 as the low layers, and considered all the other 47 layers as the high layers. For the Bert-Tiny model, we considered the first transformer block as the low layers, and considered all layers after the first transformer block as the high layers. For other models, we consider the first layer as low layers. Typically, we compute the mean distribution of interactions over 50 samples.

### I.4 DETAILS ON HOW MANY EPOCHS THE DNN WAS TRAINED BEFORE COMPUTING INTERACTIONS

To explore changes in the distribution of interactions over different orders during the overfitting phase, we trained each model for 300 epochs before calculating interactions. Specifically, we trained VGG-13 (Simonyan, 2014) on the CIFAR-10 dataset (Krizhevsky et al., 2009), VGG-16 (Simonyan, 2014) on the Tiny-ImageNet dataset (mnmoustafa, 2017), ResNet-20 (He et al., 2016) on the CIFAR-10 dataset, and the BERT-Medium model (Devlin, 2018) on the SST-2 dataset (Socher et al., 2013) to evaluate the generalization power of interactions.

To explore the relationship between hard samples and confusing samples, we trained each model for 100 epochs before calculating interactions. Specifically, we trained VGG-11 on the CIFAR-10 dataset and trained ResNet-20 on the MNIST dataset.

To explore the composition of confusing samples in different DNNs, we trained each model for 200 epochs before calculating interactions. We conducted experiments using ResNet-32 and ResNet-56 trained on the CIFAR-10 dataset.

To explore the impact of a DNN's low-layer parameters, network architecture, and high-layer parameters on the composition of confusing samples, we trained each model for 150 epochs before calculating interactions for ResNet-56 models and trained each model for 10 epochs before calculating interactions for Bert-Tiny models. We trained the ResNet-56 model on the CIFAR-10 dataset and trained the Bert-Tiny model on the SST-2 dataset.

### I.5 PREPARATION OF DISJOINT TRAINING SETS

To evaluate model behaviors under different training conditions, we constructed two disjoint training sets for each dataset under study. Specifically, for the CIFAR-10 dataset, we randomly sampled two disjoint subsets from the original training set, with each subset containing 1,000 samples. Similarly, for the SST-2 dataset, we created two disjoint subsets, each containing 2,000 samples. In both cases, the two subsets shared no overlapping samples.

In addition, we randomly sampled 200 examples from the remaining portion of the training set to serve as the testing samples for interaction computation. These testing samples were not used during training, ensuring a fair and unbiased comparison of the learned models.

We intentionally chose relatively small subset sizes (1,000 for CIFAR-10 and 2,000 for SST-2) rather than using the full training sets. The rationale behind this design choice is that large sample sizes tend to make the data distributions of the two subsets more similar, which reduces the chance of observing diverse model behaviors. By using smaller disjoint training sets, we aimed to maximize the distributional difference between the two models, thus making it easier to analyze how models trained on different data perceive sample interactions differently.

Note that the identification of confusing samples was always conducted on the testing samples, rather than on the training data, in all experiments. This further ensures the fairness of comparisons between models trained on disjoint subsets.

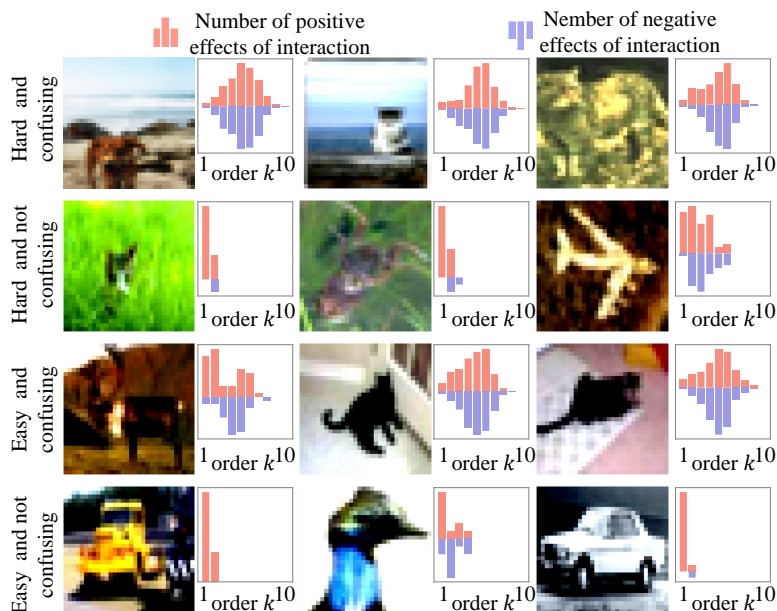

Figure 13: Confusing samples are not the same as hard samples. Some hard samples are not confusing samples, and some confusing samples are not hard samples, either.

### I.6 DETAILS ABOUT HOW TO FIND HARD SAMPLES FOR DIFFERENT DNNS

To better identify intuitively hard samples, we selected three samples from each class labeled 1 to 9 in the CIFAR-10 and MNIST datasets and reassigned their labels to 0. These samples were more likely to become hard samples, which facilitated comparison with confusing samples and enabled a more detailed analysis of their differences.

Based on this setup, we trained a DNN and recorded the loss on each sample across different training epochs. We then computed the average loss per sample and selected the top 30 samples with the highest average loss as the hard samples. Additionally, we randomly selected 30 samples from the remaining samples as easy samples for comparison.

### I.7 DETAILS ABOUT HOW TO SET TWO DNNS TO HAVE THE SAME LOW-LAYER PARAMETERS

In the experiments of Sections 3.2, we explored the impact of different low-layer parameters on the composition of confusing samples in DNNs. To compare two DNNs trained on the same datasets while ensuring they had different low-layer parameters, we replaced the low-layer parameters of the current two DNNs with those from two other well-trained DNNs that had the same architecture but different low-layer parameters. To compare two DNNs trained on different datasets while ensuring they had identical low-layer parameters, we replaced their low-layer parameters with those from a single well-trained DNN, ensuring consistency in their low-layer parameters.

## J MORE EXPERIMENTAL RESULTS

### J.1 MORE RESULTS FOR EXPLORING PARAMETERS IN HOW MANY LAYERS ARE SUFFICIENT TO DETERMINE CONFUSING SAMPLES

In this subsection, we show more results for exploring parameters on how many layers are sufficient to determine confusing samples. We compared the composition of confusing samples of two DNNs with the same $L$ low-layer parameters, which are trained on different datasets. Specifically, we conducted experiments on $L = 0, 1, 3, 5, 7, 9, 11, 13$, please see Figure 12 for details.

## J.2 MORE RESULTS FOR THE NUMBER OF INTERACTIONS EXTRACTED FROM HARD SAMPLES AND EASY SAMPLES

In this subsection, we show more results for the number of interactions extracted from hard samples and easy samples. Figure 13 shows that most hard samples encode mutually offsetting interactions, and the other type of hard samples only have a few interactions. 13 also shows that some confusing but not hard samples contain a large number of interactions, including both lots of mutually offsetting interactions and numerous non-offsetting low-order interactions. In this way, confusing samples are not the same as hard samples. Some hard samples are not confusing samples, and some confusing samples are not hard samples, either.

