# OpenReview forum: "Towards the Decisive Factor of Symbolic Generalization of DNNs"
_ICLR.cc/2026/Conference — Submitted to ICLR 2026_

### Official Review · Reviewer_A3cL · 2025-10-27

**Soundness:** 2
**Presentation:** 2
**Contribution:** 3
**Rating:** 4
**Confidence:** 4

**Summary:**

The paper identifies the cause of overfitting in symbolic generalization to be the emergence non-generalizable interactions in DNNs. Confusing samples lead to non-generalizable interactions and are distinct from samples with high loss. It then investigates the impact of different aspects of model architecture. In particular, it discovers that low-layer parameter initializations have a more pronounced effect on the average interaction order, giving rise to more confusing samples.

**Strengths:**

1) The paper takes a decisive step towards understanding symbolic generalization. In particular, it relates emergence of non-generalizable interactions to overfitting and shows clear evidence of this correlation.
2) Their observation that samples with high loss is not equivalent to confusing samples, is important and well-supported.
3) The authors also investigate the impact of inherent data quality, initializations and architectural differences on overfitting.

**Weaknesses:**

1) I'm concerned about the approach taken to discover these non-generalizable interactions by using a baseline model trained only on testing data. In particular, the interactions developed after such a training may correspond to memorizing interactions on the test data. More details on training for this baseline model (which I could not find in the appendix either) is needed to address this issue. The cited technical report He et. al. (2025) also does not talk about the training procedure.
2) The authors mention that the confusing samples were conducted on the testing samples. Can they clarify what this means? If indeed they obtain the confusing samples from a test pool, how is this relevant to the training phase?
In general, I think any claims made by conducting experiments on a test set must be very carefully made if there is any sort of training involved on the test set, due to subtleties related to data leakage or overfitting on test set.
3) There is no discussion of limitations in the main text.
4) While I believe the findings of the paper are significant from an explainability perspective, it seems difficult to transfer knowledge about confusing samples to a new dataset or a new initialization.

**Questions:**

1) The experiment on architectural difference seems to have a distinction only in terms of parameter count (Resnet-32 and Resnet-56). Renaming this section might be helpful. As a follow-up the authors can try training models without residual connections in the upper or lower or all layers.
2) The last subplot in Fig. 6(c) is missing the results for TinyBert.
3) A small suggestion to improve readability would be to summarize the plots in Figs 5-7. Could you report the average deviation from the diagonal for each plot?

---

### Official Review · Reviewer_e8C8 · 2025-10-31

**Soundness:** 2
**Presentation:** 2
**Contribution:** 2
**Rating:** 4
**Confidence:** 2

**Summary:**

This paper describes DNN overfitting as an example-level phenomenon: a small number of "perplexed examples" are introduced during training, inducing the model to learn higher-order interactions that cancel out and are not generalizable, thus increasing the training-test gap. We further find that the random initialization of low-level parameters almost determines which samples will be confused, while the impact of the data itself, high-level parameters, or architecture is relatively small. Based on these observations, we also show that suppressing nongeneralizable interactions during training can lead to small performance gains, and propose the practical implication that data valuation should vary from model to model.

**Strengths:**

1. We shift the fine-grained attribution of "overfitting/generalization" from "sample difficulty" to the chain of "low-level initialization -> confusing samples -> non-generalizable interactions", which extends the perspective of the lottery ticket hypothesis.

2, we validate our conclusions to be robust across vision /NLP, different architectures, and demonstrate practical gains from the training intervention.

**Weaknesses:**

1. The generalization test relies on the "baseline model trained on the test set": using a baseline trained only on the test set to determine whether an interaction generalizes or not can lead to the question of "evaluation bias/information leakage" in methodology. The authors need to discuss more fully the necessity of this setting and alternatives (e.g., cross-fold verification, test-free inversion criteria).

2. Universality of "only sharing low-level is aligning confusing samples": Current examples are mainly ResNet/BERT Tiny, etc. It is recommended to verify the conclusion boundary on larger models and more complex data.

3. Heuristic nature of $\eta_{avg}$: While the appendix provides evidence of correlation, it is recommended to supplement it with a more systematic sensitivity analysis (threshold τ, mask strategy, etc.) and alignment with the "true generalization failure rate".

**Questions:**

1. Confusing vs. difficult samples: Can you provide systematic metrics (e.g., intersection over union, conditional error rate improvement), rather than just graphs, to support the claim that the two are not equivalent?

2. It is mentioned in the article that "low-level initialization is the decisive factor". What is the reference for this "decisive"?

---

### Official Review · Reviewer_FDLL · 2025-11-01

**Soundness:** 3
**Presentation:** 3
**Contribution:** 3
**Rating:** 4
**Confidence:** 2

**Summary:**

This paper investigates the decisive factor that hinders symbolic generalization in deep neural networks (DNNs). Building on prior work that interprets a DNN’s output as a logical model composed of AND-OR interactions, the authors explore how networks internally form inference patterns. They introduce the concept of confusing samples, which is a small subset of training examples that lead DNNs to learn non-generalizable interactions. The key and surprising claim of the paper is that randomness in the initialization of the low-level parameters, rather than characteristics of the dataset itself, determines which samples become confusing and consequently induce non-generalizable interactions. The authors empirically demonstrate that two networks with identical architectures and similar overall performance, but different low-layer initializations, develop almost entirely different confusing sample sets. Conversely, networks sharing the same low-layer initialization exhibit much greater overlap in confusing samples, even when their high-layer configurations or training data differ. Based on these results, the paper argues that low-layer parameter initialization is the decisive factor influencing symbolic generalization in DNNs, providing new insights into the mechanisms of overfitting and generalization failure.

**Strengths:**

- **S1.** The paper introduces a novel methodological framework for identifying the cause of non-generalizable interactions in DNNs, including the concept of confusing samples.
- **S2.** By comparing confusing samples with commonly discussed hard samples, the paper provides symbolic generalization researchers with fresh perspectives and meaningful insights that challenge conventional assumptions.
- **S3.** The work effectively links symbolic generalization analysis to practical overfitting behavior, thereby offering a deeper understanding of training dynamics.
- **S4.** The ablation studies convincingly support the main claim by demonstrating that the variability in confusing samples is driven not by high-layer parameters or architectural differences, but specifically by the initialization of low-layer parameters.

**Weaknesses:**

- **W1.** The emphasis on low-layer parameter initialization is intriguing, but the experiments are limited to image classification tasks (CIFAR-10, Tiny-ImageNet) and a medium-scale BERT model on SST-2. It remains unclear whether the findings can be generalized to larger-scale architectures, other modalities (e.g., language generation), or more diverse training paradigms.
- **W2.** The definition of confusing samples depends on choices such as interaction-order thresholds and saliency measures, which can be sensitive to hyperparameters. This raises concerns about the stability and reproducibility of the concept under different configurations.
- **W3.** Extracting high-order interactions is computationally expensive, which may limit the scalability of the proposed analysis to modern, large-scale DNNs. This constraint could reduce the practical applicability of the approach.

**Questions:**

- **Q1.** To what extent does the observed phenomenon generalize to other domains and learning settings? It would be valuable for the authors to provide further discussion on its applicability beyond the current experimental scope.
- **Q2.** Are there ways to reduce the sensitivity of confusing-sample identification to hyperparameters, such as thresholds for interaction extraction, to ensure more stable and reliable characterization?
- **Q3.** In Appendix D, were the low-layer parameters kept fixed during training? If so, then the set of confusing samples would still depend on the particular initialization used. Could this mean that an unfavorable initialization might incorrectly classify important samples as confusing and lead the framework to remove them, ultimately degrading the model’s performance? Clarification from the authors on this potential risk would be valuable.

---

### Meta-Review · Area_Chair_uVw8 · 2026-01-05

**Summary:**

**Summary of Concerns** \
Reviewers identified major concerns with the **soundness and quality** of the experimental methodology. For instance, the use of a “baseline model trained on the test set” to assess whether interactions generalize raises serious concerns.  Moreover, the findings have limited **significance** due to the limited scope of the experimental analysis. It remains unclear whether the findings about “confusing samples” generalize to more realistic settings. There are also important issues with the **clarity** of the paper, as important algorithmic details and explanation about the motivation of some design choices are missing. Finally, a discussion on the limitations of the proposed analysis should be included in the paper.

**Decision** \
The paper requires a major rewriting to address the concerns highlighted by the reviewers. In its current form, the paper doesn't meet the criteria for acceptance at ICLR.

**Reviewer Concerns:**

All reviewer concerns are unaddressed

**Reviewer Scores:**

Considering that the authors didn't engage with the reviewers during the rebuttal, none of them would have increased their score.
All scores agree on the rejection of the paper.

---

### Decision · Program_Chairs · 2026-01-26

Reject